# Clinical Characteristics of Offspring Born to Parents with Type 2 Diabetes Diagnosed in Youth: Observations from TODAY

**DOI:** 10.3390/children11060630

**Published:** 2024-05-24

**Authors:** Jeanie B. Tryggestad, Megan M. Kelsey, Kimberly L. Drews, Shirley Zhou, Nancy Chang, Elia Escaname, Samuel S. Gidding, Elvira Isganaitis, Siripoom McKay, Rachana Shah, Michelle Van Name

**Affiliations:** 1Department of Pediatrics, University of Oklahoma Health Sciences Center, Oklahoma City, OK 73117, USA; 2Department of Pediatric Endocrinology, University of Colorado Anschutz Medical Campus, Children’s Hospital Colorado, Aurora, CO 80045, USA; 3Pennington Biomedical Research Center, Baton Rouge, LA 70808, USA; 4Biostatistics Center, George Washington University, Rockville, MD 20852, USA; 5Children’s Hospital of Los Angeles, Los Angeles, CA 90027, USA; 6Department of Pediatrics, UT Health San Antonio, San Antonio, TX 78229, USA; 7Geisinger Health, Danville, PA 17822, USA; 8Department of Pediatrics, Joslin Diabetes Center, Harvard Medical School, Boston, MA 02215, USA; 9Department of Pediatric Diabetes and Endocrinology, Baylor College of Medicine, Houston, TX 77030, USA; 10Division of Endocrinology and Diabetes, Children’s Hospital of Philadelphia, Philadelphia, PA 19104, USA; 11Department of Endocrinology, Yale School of Medicine, New Haven, CT 06510, USA

**Keywords:** diabetes, medication use, recidivism, perinatal complications, obesity

## Abstract

Diabetes exposure during pregnancy affects health outcomes in offspring; however, little is known about in utero exposure to preexisting parental youth-onset type 2 diabetes. Offspring born to participants during the Treatment Options for Type 2 Diabetes in Adolescent and Youth (TODAY) study were administered a questionnaire at the end of the study. Of 457 participants, 37% of women and 18% of men reported 228 offspring, 80% from female participants. TODAY mothers had lower household income (<$25,000) compared to TODAY fathers (69.4% vs. 37.9%, *p* = 0.0002). At 4.5 years of age (range 0–18 years), 16.7% of offspring were overweight according to the parental report of their primary care provider, with no sex difference. Offspring of TODAY mothers reported more daily medication use compared to TODAY fathers (50/183, 27.7% vs. 6/46, 12.2%, [*p* = 0.04]), a marker of overall health. TODAY mothers also reported higher rates of recidivism (13/94) than TODAY fathers (0/23). An Individualized Education Plan was reported in 20/94 (21.3%) offspring of TODAY mothers compared to 2/23 (8.7%) of TODAY fathers. This descriptive study, limited by parental self-reports, indicated offspring of participants in TODAY experience significant socioeconomic disadvantages, which, when combined with in utero diabetes exposure, may increase their risk of health and educational disparities.

## 1. Introduction

Diabetes diagnosed either before or during pregnancy is associated with morbidity in the mother and morbidity and mortality in the offspring. In the Treatment Options for Type 2 Diabetes in Adolescents and Youth (TODAY) study, 141 of the 452 female participants with youth-onset type 2 diabetes (T2D) reported 260 pregnancies [1], with high rates of perinatal complications and congenital anomalies in the offspring [1]. The risk for congenital malformations in infants born to mothers with T2D have been reported to be between 2 and 6% [2,3,4]; however, in the TODAY cohort, rates were much higher but similar to the Next Generation cohort [5]. Specifically, of 179 live births in TODAY, 18 offspring (10%) had a cardiac anomaly and 18 (10%) had another form of congenital anomaly [1]. While we have previously published pregnancy complications and perinatal outcomes in the women in TODAY, offspring outcomes from the men in TODAY and longer-term offspring outcomes have not yet been reported. 

Exposure to pregestational or gestational diabetes (GDM) has significant effects on indices of metabolic health during the first year of life that can persist into early childhood. Exposure to GDM has been associated with increased adiposity by 1 year, persisting to ages 4 and 7 years and resulting in the risk of metabolic syndrome by age 11 years [6,7]. Many other studies have shown that exposure to GDM is associated with increased BMI, waist circumference, and adiposity [8,9,10,11]. In a cohort of First Nations women with youth-onset diabetes, almost 90% of the youth aged 2–19 were overweight or obese [12,13]. In siblings born before and after a maternal diagnosis of T2D, the BMI of the sibling exposed to T2D in utero was 2.6 kg/m2 higher than the sibling born before the T2D diagnosis [14]. However, in the PANDORA study, the offspring exposed to GDM or T2D had lower BMI in infancy than peers not exposed to diabetes in utero, with children exposed to T2D having lower mean peak BMI [15]. Thus, exposure to diabetes in utero has long-standing impacts on offspring adiposity and metabolic health.

Exposure to diabetes in utero may also have a lasting impact on neurocognitive development in offspring. Verbal IQ and motor development appear to be adversely impacted by in utero exposure to diabetes [16,17,18]; as well, cognitive ability in offspring may be inversely related to maternal glycemia [19,20]. Specifically with youth born to mothers with T2D or GDM, 23% of an Australian cohort were at least 2SD below the mean in one developmental domain [21]. These studies suggest that diabetes exposure in the intrauterine environment has an impact on cognitive and neurologic development. While it is clear that the maternal intrauterine environmental exposure to diabetes impacts the neurocognitive development of the offspring, it is not clear what impact paternal diabetes has on neurocognitive development.

While it is clear that the diabetic milieu of pregnancy impacts offspring metabolic and neurocognitive development, most of the literature has focused on exposure to GDM. However, the specific effects on infants born to mothers with pre-existing youth-onset T2D and, particularly, outcomes in the infants born to males with youth-onset T2D have not been explored. The objective of this analysis was to explore the impact of parental diabetes on perinatal complications and congenital defects, as well as longer-term health and developmental outcomes in offspring of TODAY (average age 4.5 years (range 0–18 years, IQR [2,6])), specifically comparing infants born to mothers with youth-onset T2D (TODAY mothers) to infants born to fathers with youth-onset T2D (TODAY fathers).

## 2. Materials and Methods

The TODAY study was described previously [22]. Briefly, 699 participants, 10–17 years old, diagnosed with T2D by American Diabetes Association (ADA) criteria [23], with duration 2 years or less, BMI ≥ 85th percentile for age and sex, and absence of pancreatic autoantibodies, were enrolled across 15 clinical centers in the United States [24]. Participants were randomized to metformin alone, metformin with rosiglitazone, or metformin plus a lifestyle intervention, then followed for 2–6 years [24]. After the clinical trial (2004–2011), an observational follow-up study (TODAY2) was conducted in two phases (Phase 1 2011–2014, Phase 2 2014–2020) [25]. During the first phase, participants received standard diabetes care. In the second phase, clinical care was no longer provided through the study, but annual visits continued for the collection of biologic specimens, assessment of microvascular and macrovascular complications, and capture of demographic and health information data.

For female participants, pregnancy data, including maternal and fetal information, were obtained prospectively every 3–6 months in TODAY and annually in TODAY2. Additionally, participants gave consent to obtain records for the pregnancy, delivery, and neonate, thus allowing review and abstraction of the data. In the case where there was a discrepancy between what the subject reported and the reviewed records, the information from the medical record was used in the analysis. All study procedures were carried out in accordance with The Code of Ethics of The World Medical Association and approved by each clinical center IRB. Written informed consent was obtained prior from all participants and guardians as necessary.

For the last clinical visit of TODAY2, questionnaires were developed and administered to collect information regarding the offspring of all participants. The questionnaires were used to gather data regarding the birth of the infant and any complications after delivery. Specifically, all participants were asked to complete one form for every living child regarding the child’s age and weight status (normal weight, underweight, overweight as described by the children healthcare provider) as a marker of metabolic health. Additionally, the necessity for medical specialty care required and medications used by the child were used to assess overall health burden. Information regarding school information, such as individualized learning plan (IEP) and recidivism at any grade level, if applicable, was used to assess neurocognitive development. Finally, the participants were asked with whom the child lived to determine the primary care provider for the child. For TODAY mothers, birth records were obtained from record review for the offspring from consenting participants serving as the source of information for pregnancy outcomes, and the questions about pregnancy, delivery, and neonatal outcomes were not asked. However, the questions relating to pregnancy were asked for the few females who had not consented for access to medical records previously. The responses to the questionnaire were the primary source of information for all offspring of TODAY fathers. No medical records for the pregnancy or birth were obtained for offspring of the male participants. All post-delivery information on TODAY offspring was collected through the questionnaire. 

Descriptive statistics reported are means and standard deviations for continuous variables or frequencies and percentages for categorical variables. Two-sided *t*-tests were utilized to conduct comparisons for continuous variables, whereas χ^2^-test, Fischer’s exact tests, and logistic regression were used for comparison of the frequencies of non-continuous outcomes. *p*-values < 0.05 were considered statistically significant. The analysis presented is exploratory, thus, no adjustments were made for multiple testing. All analyses were conducted using SAS 9.4 (SAS Institute, Cary, NC, USA). 

## 3. Results

### 3.1. Description of the Cohort

A total of 457 participants completed the offspring questionnaires (65% of the original cohort). The analysis cohort was more non-Hispanic white with a slight reduction in Hispanics for both sexes (Appendix A) compared to the full TODAY cohort. Additionally, the females who remained for the final TODAY2 study visit were from families with lower household income and were younger at the time of enrollment (Appendix A). No other differences were noted between those who completed the questionnaire vs. not. Of the TODAY mothers, 111 out of a total of 299 (37%) reported at least one offspring. Of the TODAY fathers, 29 out of a total of 158 (18%) reported at least one offspring. TODAY fathers were older than TODAY mothers (*p* = 0.007), likely reflecting older age at study entry in the overall cohort for males [26]. Race and ethnicity differed between TODAY mothers compared to TODAY fathers. TODAY mothers were more likely to identify as non-Hispanic Black, while TODAY fathers were more likely to identify as Hispanic (*p* = 0.016). TODAY mothers were more likely to have a household income below $25,000 compared to TODAY fathers (*p* < 0.001, Table 1). As age, race/ethnicity, and income at the final visit differed between TODAY mothers and fathers, adjustments for these variables were made in subsequent models. Based on the report of with whom the child lived, 92% of TODAY mothers and 60% of TODAY fathers reported being a primary caregiver (Appendix A).

### 3.2. Perinatal Complications

The participants reported a total of 228 offspring, 80% of which were offspring of the TODAY mothers. Preterm delivery was more common in offspring of TODAY mothers compared to TODAY fathers (58% vs. 15%, *p* < 0.001, Table 2). No differences were observed in the rate of neonatal hypoglycemia after adjusting for race/ethnicity, income, and age of the participants at the final visit. The rates of respiratory distress at delivery were higher in offspring of TODAY fathers, bordering on significance in the unadjusted models, but were no longer significant after adjustment. When considering pre-term deliveries, respiratory distress was higher in the infants born preterm to mothers with diabetes, but this association was not found in the infants born to males related to small sample size. Cardiac anomalies tend to be higher in the infants of TODAY mothers, although not statistically significant without or with adjustment. No differences were observed in other anomalies. 

### 3.3. Long-Term Health Outcomes

The average age of the offspring was 4.5 years (range 0–18 years) at the time of the survey, with equal numbers of males and females. Overall, 17% of the offspring of TODAY mothers and 15% born to TODAY fathers were reported to be overweight per their healthcare provider, with similar results across offspring age groups (Table 3). Results were adjusted for participant race/ethnicity, income, and age at final study visit excluding missing data, but no differences in weight category were observed after the adjustment.

With regard to overall health, TODAY mothers were more likely to report that their offspring saw a specialist (26.1%) than TODAY fathers (11.6%, *p* = 0.04); however, after adjusting for participant race/ethnicity, income, and age at final study visit, the difference was no longer significant. TODAY mothers were also more likely to report medication use on a regular basis by their offspring (27.7% compared to 12.2% in TODAY fathers), which remained significant after adjustments (*p* = 0.0495, Table 4).

### 3.4. Developmental Outcomes

Of the 228 TODAY offspring, 53% of the offspring of TODAY mothers and 52.3% of the offspring of TODAY fathers were reported to be attending school. Regarding neurocognitive development in those attending school, 13.8% of the offspring of TODAY mothers repeated a grade in school, while no offspring of TODAY fathers repeated a grade, though this was not statistically significant (*p* = 0.20 after adjusting for participant race/ethnicity, income, and age at final study visit). An IEP was reported in school-age offspring in 21.3% TODAY mothers and in 8.7% of TODAY fathers (*p* = 0.11 after adjustment, Table 4).

## 4. Discussion

Studies demonstrating specific effects of youth-onset T2D on the health and development of their offspring are very limited. The 15-year duration of follow-ups in the TODAY study offered a unique opportunity to address these topics. Higher rates of prematurity and medication use were observed in offspring of TODAY mothers with a trend toward higher rates of cardiac anomalies and specialized care needs. While intrauterine exposure to the diabetic milieu during gestation as a result of maternal youth-onset T2D did appear to have some effect on overall health outcomes, it did not appear to have any additional effect on BMI or measures of school performance, as offspring of TODAY mothers and fathers had similar rates of overweight, recidivism, and IEP by report. It should be noted that very few of the offspring were at the typical age of diagnosis for youth-onset T2D (N = 6 were 12 years or older), so it is too soon to understand the impact of young-onset T2D on diabetes development in the offspring.

The rates of preterm delivery in women with diabetes exceed those of women with normal glycemia. From the Maternal–Fetal Medicine Units Network for the National Institutes of Health cohort, women with diabetes were more likely to deliver before 37 weeks (38%) compared to those without diabetes (13.9%) [27]. Diabetes during pregnancy, especially pregestational diabetes treated with insulin, is associated with increased rates of preeclampsia necessitating an early delivery. This likely contributed to the high rates of preterm deliveries in the infants born to female participants (58% delivering before 37 weeks) in TODAY. The presence of diabetes or other maternal complications was not assessed in the mothers of offspring born to TODAY fathers. 

Rates of respiratory distress in infants are highly linked to prematurity [28]. While the rates of respiratory distress tended higher in the infants born to TODAY fathers, this did not persist after adjustment. This is related to the very small sample size in the males, with a significant number reporting they were unsure of any respiratory distress at birth. When only considering the infants born to mothers with diabetes, respiratory distress was higher in those infants born before 37 weeks as might be expected. Further exploration in a larger cohort will be necessary to identify factors associated with respiratory distress in infants born to males with youth-onset T2D. The lack of difference in neonatal hypoglycemia between offspring of TODAY mothers and fathers is similarly surprising due to the known impact of inadequately controlled diabetes in utero on risk for neonatal hypoglycemia. However, the lack of differences may be explained by reporting bias in TODAY fathers and/or lack of an adequate sample size to detect differences. 

The rates of cardiac anomalies tended to be higher in the infants born to TODAY mothers compared to TODAY fathers, with no difference in other anomalies. Based on a meta-analysis, the risk of any congenital heart malformation was 3.8-fold in infants born to mothers with pregestational diabetes compared to those without diabetes [29]. Offspring of TODAY mothers with worse glycemic control, defined as HbA1c ≥ 8% during pregnancy, had higher rates of cardiac anomalies [1]. Therefore, exposure to hyperglycemia in utero may be a potential mediator of congenital heart disease risk; however, this has not been consistent across studies [30]. 

The reported rates of overweight in TODAY (17% for all age ranges) were surprisingly similar to national rates reported for youths aged 2–19 years in the National Health and Nutrition Examination Survey (NHANES) from 2017–2018 (16%, 19.3%, and 6.1% for overweight, obesity, and severe obesity, respectively), even in the lowest socioeconomic group [31]. However, the observed rates of overweight within the specific 2–5 years and 6–11 years age categories (16.7% and 23.1%, respectively) were slightly higher than the reported obesity rates in NHANES (13.4% and 20.3%, respectively) [31]. It is important to note that parents’ weight perceptions may be inaccurate [32], resulting in mis-categorization. The findings of the Exploring Perinatal Outcomes among Children (EPOCH) study found that offspring BMI was not significantly impacted by diabetes exposure in utero until after age 26 months [33]. Given that the average age of the cohort was 4.5 years, it is possible the assessments were too early to capture effects on offspring obesity. From the Pregnancy and Neonatal Diabetes Outcomes in Remote Australia (PANDORA) study, exposure to diabetes early in pregnancy was associated with increased adiposity in the infants [34]. Additionally, early changes in adiposity independent of BMI have been noted in cohorts exposed to diabetes in utero [35], which may have more prognostication than BMI alone for future metabolic disease. In addition to the impact of diabetes exposure, both maternal [36] and paternal [37] obesity have been associated with greater childhood adiposity in the offspring. Thus, the risk of excess adiposity is higher in TODAY offspring for multiple reasons, and accurate assessment of BMI as well as adiposity is needed to better understand the specific influence of parental youth-onset T2D on offspring adiposity.

TODAY offspring, particularly offspring of TODAY mothers, also displayed higher-than-average overall health concerns. TODAY mothers reported chronic medication use in 28% of offspring, which is greater than that reported in NHANES (18%) in children aged 0–11 years [38]. TODAY fathers reported similar chronic medication use to NHANES; however, 10.9% did not know anything about medication usage in their offspring. The offspring of TODAY mothers also reported a higher rate of subspecialty physician visits (26.1%) compared to the average rate among children in the US (13% per the National Survey of Children’s Health), comparable to children with special healthcare needs, 34% of whom see a subspecialist [39]. These subspecialty visits may be in part related to high rates of congenital anomalies, present in 20% of this group. Thus, the increased medication usage and trend for more specialty visits in offspring of TODAY mothers may be related to overall worse health outcomes due to in utero exposure to diabetes and its resulting complications, such as prematurity and congenital anomalies. 

Data suggest that in utero exposure to diabetes may have detrimental neurocognitive effects on offspring, with reports of lower IQ, increased risk of autism spectrum disorders, and, possibly, attention deficit/hyperactivity disorder in a recent meta-analysis [40]. The offspring of TODAY mothers reported an IEP in 21.3% of children, more than double the rate reported by the National Health Interview Survey in 2018 of 9% [41]. Rates reported in offspring of TODAY fathers were similar to national data. Frequency of recidivism, at 13.8%, was also higher in offspring of TODAY mothers compared to the national rate of 6.5% from the Data Resource Center for Child & Adolescent Health, including 6–17-year-old youths [39]. While the number of school-age offspring in TODAY is small, the data suggest higher rates of learning challenges in TODAY mothers compared to fathers, suggesting that the in utero exposure to the diabetic milieu may impact neurocognition as seen in previous studies. 

This study has several strengths. Data were able to be collected from 228 offspring of parents with youth-onset T2D diabetes, one of the largest groups with this unique exposure. The parents from TODAY were studied over an average of 13 years, allowing for a cross-sectional examination of the impact of youth-onset T2D on the participants’ offspring. 

Some limitations are acknowledged as well. Fewer male participants in TODAY reported a child than female participants in TODAY, and overall event numbers were low, impacting the power of this study. Also, fewer fathers reported being the primary caregiver for the children, which may be a source of bias in the data collection. While the offspring birth data were collected from record review in the offspring of TODAY mothers, all birth outcomes for the offspring of TODAY fathers were obtained through recall. Also, no information was collected regarding the mother of the infants born to TODAY fathers. As only 60% of the fathers reported themselves as the primary caregivers, the risk of recall bias and missing data was high. Importantly, all information on metabolic health and educational status in the offspring were obtained through recall, thus, important anthropometrics such as adiposity were not able to be defined. Another significant limitation is the inability to tease out the impact of socioeconomic status on outcomes, especially regarding those requiring an IEP or repeating a grade. While this again increases the risk of bias, it lays the foundation for future studies to better explore the impact of parental youth-onset T2D in metabolic and neurocognitive development of their offspring. We were also limited due to a lack of genomic analysis in the cohort. We also recognize that many additional aspects of overall health, including nutrition and physical activity, are linked with childhood BMI but were not collected in this study. 

## 5. Conclusions

In conclusion, offspring of females with youth-onset T2D reported higher rates of health and learning concerns not only compared to the offspring of the males in TODAY, but also compared to national population data. The difference in outcomes based on the sex of the parent with youth-onset T2D is likely related to effects of the intrauterine environment rather than the genetic or epigenetic contributions of the parent. This descriptive study, although limited by parental self-report, indicates that the offspring of participants in the TODAY study experience significant socioeconomic disadvantages which, when combined with the additional risks associated with exposure to diabetes in utero, may put them at greater risk of health and educational disparities. The implications of these findings would suggest that infants born to parents with diabetes may need early interventions targeting risk for obesity and neuropsychiatric evaluation for appropriate school accommodations. Future studies geared specifically at examining the specific influence of maternal as well as paternal youth-onset T2D diabetes in a prospective manner are needed to better understand the generational cardiometabolic and neurocognitive impacts of this disease, and to better delineate additional effects of in utero exposure. 

## Figures and Tables

**Table 1 children-11-00630-t001:** Descriptive table for participants who completed an offspring questionnaire by sex and offspring status.

	Females		Males		*p*-Value *
	No Offspring (N = 188)	≥1 Offspring (N = 111)	No Offspring (N = 129)	≥1 Offspring (N = 29)	
Age in years (mean, SD) at final visit	26.1 (2.5)	27.0 (2.4)	27.1 (2.4)	28.3 (2.0)	0.0069
Duration of T2D in years (mean, SD) at final visit	13.5 (1.5)	13.7 (1.5)	13.6 (1.5)	13.9 (1.3)	0.5328
Race/Ethnicity (%)					0.0161
White, non-Hispanic	19.1%	18.0%	24.0%	10.3%	
Black, non-Hispanic	37.8%	41.4%	30.2%	24.1%	
Hispanic	37.8%	27.0%	41.9%	58.6%	
Other	5.3%	13.5%	3.9%	6.9%	
Participant Income at final visit (%)					0.0002
<$25,000	55.9%	69.4%	56.6%	37.9%	
$25,000–$49,999	33.5%	20.7%	28.7%	37.9%	
>$50,000	6.9%	2.7%	11.6%	20.7%	
Unknown	3.7%	7.2%	3.1%	3.4%	
Participant education at final visit (%)					0.8626 **
Less than high school	6.4%	18.0%	10.9%	20.7%	
High school degree or equivalent	61.2%	66.7%	59.7%	69.0%	
Some college	12.8%	6.3%	11.6%	6.9%	
College degree or higher	19.7%	9.0%	17.8%	3.4%	
Parental diabetes (%)	72.9%	75.7%	74.4%	86.2%	0.2239
Loss of glycemic control during clinical trial (%)	42.0%	52.3%	48.8%	55.2%	0.7791
Time-weighted A1c (mean, SD) at final visit	8.2 (2.1)	8.8 (1.9)	8.6 (2.2)	8.7 (2.3)	0.8140
Time weighted BMI in kg/m^2^ (mean, SD) at final visit	37.0 (8.3)	35.8 (6.6)	36.2 (8.1)	34.6 (6.7)	0.3891

* *p*-value compares the characteristics of the males and females with ≥1 offspring. ** Indicates testing using Fisher’s exact test.

**Table 2 children-11-00630-t002:** Delivery and perinatal complications and congenital defects by participant sex and overall.

	Participant Sex		Unadjusted *p*-Value	Adjusted *p*-Value
	Female (Mothers)	Male (Fathers)		
Number of offspring	182	46		
Preterm Delivery (%)	58.0%	15.2%	<0.0001	<0.0001
Respiratory Distress (%)	21.6%	50.0%	0.0541 *	0.2465
Neonatal hypoglycemia (%)	34.6%	30.0%	1.0000 *	0.6370
Cardiac Anomalies (%)	12.5%	2.2%	0.0521 *	0.0630
Other anomalies (%)	11.4%	8.9%	0.7902 *	1.0000 *

*p*-value compares the outcomes for the females (mothers) and the males (fathers). Adjustments are participant race/ethnicity, income and age at final study visit. Percentages represent the proportion present among the known outcomes (i.e., excludes responses of unknown or don’t know). * Indicates testing using exact tests.

**Table 3 children-11-00630-t003:** Kids weight status table by participant sex and overall.

	Participant Sex		Unadjusted *p*-Value	Adjusted *p*-Value
	Female (Mothers)	Male (Fathers)		
Current child weight category (overall) (N)	182	46	0.1980	0.1884
Normal (%)	65.9%	76.1%		
Underweight (%)	9.9%	2.2%		
Overweight (%)	17.0%	15.2%		
Don’t know/missing (%)	7.1%	6.5%		
Current child weight category (age < 2) (N)	42	10	0.1870 *	**
Normal (%)	64.3%	90.0%		
Underweight (%)	21.4%	0.0%		
Overweight (%)	9.5%	0.0%		
Don’t know/missing (%)	4.8%	10.0%		
Current child weight category (age 2–5) (N)	84	16	0.6020 *	0.8779
Normal (%)	69.0%	75.0%		
Underweight (%)	6.0%	0.0%		
Overweight (%)	16.7%	25.0%		
Don’t know/missing (%)	8.3%	0.0%		
Current child weight category (age 6–11) (N)	52	18	0.8929	0.3203
Normal (%)	61.5%	66.7%		
Underweight (%)	7.7%	5.6%		
Overweight (%)	23.1%	16.7%		
Don’t know/missing (%)	7.7%	11.1%		

*p*-value compares the offspring by parental sex. Adjustments are participant race/ethnicity, income and age at final study visit. Don’t know/missing not included in the models. * Comparisons are by Fisher’s exact test. ** Model does not converge, no test possible.

**Table 4 children-11-00630-t004:** Childhood development table by participant sex and overall.

	Participant Sex		Unadjusted *p*-Value **	Adjusted *p*-Value **
	Female (Mothers)	Male (Fathers)		
N (Number of offspring)	182	46		
Current age (years) (mean ± SD)	4.4 ± 3.2	4.8 ± 3.5	0.4364	0.8180
Attend school (%)	53.0%	52.3%	0.9273	0.7934
Repeated grade (%) *	13.8%	0.0%	0.0691 ^#^	0.1964 ^#^
IEP present (%) *	21.3%	8.7%	0.2374 ^#^	0.1065 ^#^
See specialist (%) ^§^	26.1%	11.6%	0.0436	0.1530
Take medication (%) ^¥^	27.7%	12.2%	0.0384	0.0495

* Repeated grade, IEP present only includes those who attended school. ** *p*-value compares the offspring characteristics by parental sex. Adjustments are participant race/ethnicity, income and age at final study visit. ^#^ Comparisons are by exact test. ^§^ Specialties include cardiology, pulmonary, gastrointestinal, endocrinology, nephropathy, urology, and psychology. ^¥^ Medication medical conditions included asthma, heart conditions, attention issues, seizure disorders, stomach and urinary problems.

## Data Availability

Data from the Treatment Options for Type 2 Diabetes in Adolescents & Youth Long Term Follow-Up [(V1)/https://doi.org/10.58020/z6n1-wc73 reported here are available upon request at the NIDDK Central Repository (NIDDK-CR) website, Resources for Research (R4R), https://repository.niddk.nih.gov/.

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
