# Peer review of "Clinical Characteristics of Offspring Born to Parents with Type 2 Diabetes Diagnosed in Youth: Observations from TODAY"

_children, 2024, doi:10.3390/children11060630_

Round 1

Reviewer 1 Report

Comments and Suggestions for Authors

The authors have presented the Clinical Characteristics of Offspring Born to Parents with Type 2 Diabetes Diagnosed in Youth (Observations from TODAY). I find a little disturbing the high percentage of plagiarism therefore i suggest authors to reconsider some words associations and rewrite them.

Also, please add a citation to ADA in the methodology section and also after the TODAY studies.

There are no comments about nutritional evaluation and family food behaviors. We know that a child learns food behavior from their parents. If parents are obese, most likely their child will develop obesity. Especially nowadays when children are not so active like several years ago when technology and game apps were not so available. And this obesity leads to several medical conditions.

The association between youth diabetes in mothers and clinical characteristics in their offsprings is weak. However the study is interesting and needs to be validated.

Author Response

The authors have presented the Clinical Characteristics of Offspring Born to Parents with Type 2 Diabetes Diagnosed in Youth (Observations from TODAY). I find a little disturbing the high percentage of plagiarism therefore i suggest authors to reconsider some words associations and rewrite them.

We acknowledge the high similarity rate, but we would like to request additional considerations for sources numbers 1-3 in the originality report. The majority of the similarity in source number 1 is related to our Appendix A which includes all the clinical centers and institutions involved in the multi-site, multi-year study. We feel very strongly that all the involved people should be recognized for their efforts, thus the reason for the similarity is this is a uniform list which has been included with all publications to give appropriate credit. Regarding source number 2, the similarity is again in the acknowledgements. As we need to make the same acknowledgments for every manuscript, there will be repetition, but we again feel that all contributions and our funding sources should be acknowledged. With regard to source number 3, this is a published abstract for a presentation at Pediatric Endocrine Society meeting in 2022 for which the proceedings were published in Hormone Research. As this is the manuscript of that work, there will be similarities. We apologize for not pointing this out in the cover letter.

 Since source number 4 was the next largest area of similarity, we have attempted to revise to decrease the similarity. We hope that the revisions along with the explanation will meet the journal requirements.

Also, please add a citation to ADA in the methodology section and also after the TODAY studies.

We have added the citation for the ADA standards of care from 2003 which would have been the criteria at the time that TODAY started. We have added the TODAY study citations at each phase to further clarify.

There are no comments about nutritional evaluation and family food behaviors. We know that a child learns food behavior from their parents. If parents are obese, most likely their child will develop obesity. Especially nowadays when children are not so active like several years ago when technology and game apps were not so available. And this obesity leads to several medical conditions.

We agree with the reviewer that food behaviors and physical activity are critical mediators of childhood BMI. Unfortunately, this information was not collected in the present study, but it is a significant area that will be considered in future studies. We have included this as a limitation of the present study.

The association between youth diabetes in mothers and clinical characteristics in their offsprings is weak. However the study is interesting and needs to be validated.

We thank the reviewer for the comment. We acknowledge that the only significant difference in clinical characteristics is preterm delivery and medication use in comparing the infants born to mothers with diabetes to infants born to fathers with diabetes. This is largely driven by the small number of infants born to fathers. This is cited in the limitations.

Reviewer 2 Report

Comments and Suggestions for Authors

This study is a highly valuable investigation that was able to follow the characteristics of children of patients with youth-onset type 2 diabetes who tested negative for pancreatic autoantibodies. Its findings revealed that descendants of diabetic patients may require medical or social support from a very early stage of life. The patient registration process is accurately described, and the statistical analysis has been conducted correctly. While acknowledging the small sample size, it is appropriately stated as a limitation.  

Minor point

-line 86, as the abbreviation ADA is first introduced, please state the full name: American Diabetes Association.

Author Response

This study is a highly valuable investigation that was able to follow the characteristics of children of patients with youth-onset type 2 diabetes who tested negative for pancreatic autoantibodies. Its findings revealed that descendants of diabetic patients may require medical or social support from a very early stage of life. The patient registration process is accurately described, and the statistical analysis has been conducted correctly. While acknowledging the small sample size, it is appropriately stated as a limitation.  

We thank the reviewer for the careful review.

Minor point

-line 86, as the abbreviation ADA is first introduced, please state the full name: American Diabetes Association.

Thank you for pointing this out. We have now state the full name.

Round 2

Reviewer 1 Report

Comments and Suggestions for Authors

The authors have modified and improved the manuscript as suggested.